# Circulatory miRNA as a Biomarker for Therapy Response and Disease-Free Survival in Hepatocellular Carcinoma

**DOI:** 10.3390/cancers12102810

**Published:** 2020-09-29

**Authors:** Muhammad Yogi Pratama, Alessia Visintin, Lory Saveria Crocè, Claudio Tiribelli, Devis Pascut

**Affiliations:** 1Fondazione Italiana Fegato-ONLUS, Area Science Park, ss14, km163.5, 34149 Trieste, Italy; lcroce@units.it (L.S.C.); ctliver@fegato.it (C.T.); devis.pascut@fegato.it (D.P.); 2Faculty of Medicine, Universitas Hasanuddin, Makassar 90245, Indonesia; 3Dipartemento di Scienze della Vita, Università degli Studi di Trieste, 34149 Trieste, Italy; 4Department of Medical Sciences, University of Trieste, 34149 Trieste, Italy; alessia.visintin@units.it; 5Clinica Patologie Fegato, Azienda Sanitaria Universitaria Integrata di Trieste (ASUITS), Via Giovanni Sai 7, 34149 Trieste, Italy

**Keywords:** non-coding RNA, serum biomarker, TACE, radiofrequency ablation, liver tumor

## Abstract

**Simple Summary:**

In the present study, we identified new prognostic serum miRNA biomarkers to predict therapy response and disease-free survival according to different therapeutic strategies in hepatocellular carcinoma (HCC). MiR-4443, miR-4454, and miR-4530 were significantly related to treatment response to HCC curative treatments (liver resection and radiofrequency ablation) while miR-4492 was related to treatment response of trans-arterial chemoembolization (TACE). The value of these results can contribute to setting the basis for personalized medicine for HCC patients and to improve the survival of HCC patients in the future.

**Abstract:**

The clinical outcome of hepatocellular carcinoma (HCC) treatment remains unsatisfactory, contributing to the high mortality of HCC worldwide. Circulating miRNAs have the potential to be a predictor of therapy response. Microarray profiling was performed in serum samples of 20 HCC patients before treatment. Circulating miRNAs associated with treatment response were validated in 86 serum HCC samples using the qRT-PCR system. Patients were treated either with curative treatments (resection or radiofrequency) or trans-arterial chemoembolization (TACE), and grouped according to therapy response in complete responders (CR) and partial responders or progressive disease (PRPD), following mRECIST criteria. Four miRNA candidates from the discovery phase (miR-4443, miR-4454, miR-4492, and miR-4530) were validated. Before therapy, miR-4454 and miR-4530 were up-regulated in CR to curative treatments (2.83 fold, *p* = 0.02 and 2.33 fold, *p* = 0.008, respectively) and were able to differentiate CR from PRPD (area under the curve (AUC) = 0.74, sens/spec 79/63% and AUC = 0.77, sens/spec 72/73%). On the contrary, miR-4443 was 1.95 times down-regulated in CR (*p* = 0.05) with an AUC of 0.72 (sens = 70%, spec = 60%) in distinguishing CR vs. PRPD. The combination of the three miRNAs was able to predict the response to curative treatment with an AUC of 0.84 (sens = 72%, spec = 75%). The higher levels of miR-4454 and miR-4530 in were associated to longer overall survival (HR = 2.79, *p* = 0.029 and HR = 2.97, *p* = 0.011, respectively). Before TACE, miR-4492 was significantly up-regulated in CR patients (FC = 2.67, *p* = 0.01) and able to differentiate CR from PRPD (AUC = 0.84, sens/spec 84.6/71%). We demonstrated that different miRNAs predictors can be used as potential prognostic circulating biomarkers according to the selected treatment for HCC.

## 1. Introduction

Hepatocellular carcinoma (HCC) is the most common type of primary liver cancer. It represents the seventh most frequent cancer and the fourth leading cause of cancer-related death worldwide in 2018 [1]. A multitude of etiological risk factors such as hepatitis B and C, alcohol consumption, non-alcoholic fatty liver disease (NAFLD), obesity, metabolic syndrome, dietary exposure to aflatoxin B1, or tobacco abuse were linked to the HCC global incidence [2,3,4,5,6]

Currently, HCC remains as one of the most “difficult-to-treat” cancers due to the late diagnosis and its poor prognosis rate. Indeed, the vast majority of HCC are diagnosed at an advanced stage and no longer eligible for surgical approaches or liver transplantation [7]. As a matter of fact, 80–90% of newly diagnosed HCC are not considered eligible for any curative treatments [8]. Moreover, the outcome of such treatment remains unsatisfactory. Generally, resection has a satisfactory overall (OS) and disease-free survival rate (DFS) only in patients at early stages [9,10]. Indeed, the reported OS and five-year DFS are 38.5–53.0% and 29.4–34.2%, respectively [9,11]. Non-curative therapies, such as trans-arterial chemoembolization (TACE), that remains as the recommended first-line therapy for BCLC B grade [12], had a poor long-term outcome [13]. The partial response was reported only in 15–55% of patients, with the reported five-year survival rates of TACE only ranged from 1–8% [14,15]. BLCLC staging and current treatment algorithms divide patients in a few subgroups, without considering intrinsic molecular aspects of the tumor. Thus, the presence of predictive algorithms might provide clinicians with a new tool for patient selection and prioritization. In this regard, circulating biomarkers, with the benefit of being non-invasive, easy to measure and cost-effective, can be considered of new importance.

Among the potential biomarkers, highly conserved 18–25 nt non-coding RNAs, microRNAs (miRNAs), remain promising due to their fundamental role in the regulation of various key pathways in HCC [16,17]. In addition, circulating miRNAs found in blood and serum are remarkably stable from degradation [18,19], making them suitable as predictive non-invasive prognostic biomarkers in clinical settings. In the present study, we identified new prognostic and predictive miRNA biomarkers for therapy response and disease-free survival according to different therapeutic strategies. These new tools can contribute to setting the basis for personalized medicine for HCC patients.

## 2. Results

### 2.1. Characteristics of the Study Population

The demographic features of the groups are shown in Appendix A. Among the 86 HCC patients enrolled in the study, 40 patients received curative treatments, radiofrequency ablation or resection, while 46 patients underwent TACE procedures. Following curative treatments, 27 had a complete response (CR) while 13 had a partial response or progressive disease (PRPD). Fourteen patients completely responded to TACE while 32 were non-responders. Participants were predominantly male (80.23%) with a mean age of 72.4 (*p* < 0.01). The most frequent etiologies of chronic liver disease were alcohol abuse with or without the metabolic disease (53.49%), and viral hepatitis (39.53%). Most of the patients were Child-Pugh score A (78.05%), whereas it was B for the remaining with a substantial conserved liver function. BCLC 0/A and B/C scores were recorded in 74.42% and 27.90% of the patients, respectively (*p* < 0.001). Patients with BCLC score B or C were predominantly considered eligible for TACE according to BCLC treatment guidelines. When considering the BCLC stages in TACE-treated patients, we observed no statistically significant differences in the distribution of BCLC stages among the two categories (CR vs. PRPD), chi-square statistic= 1.41 and *p* value 0.23. 

With respect to tumor mass and the number of lesions, 11.62% of subjects showed a single nodule smaller or equal to 2 cm in diameter, 52.33% had either a single nodule ranging 2 to 5 cm, or 2–3 nodules up to 3 cm each, while 34.88% had single nodule larger than 5 cm in diameter or multifocal. As for AFP level, 56.97% had AFP level lower than 20 ng/mL, 13.95% between 20 ng/mL and 400 ng/mL, and 6.98% with AFP level above 400 ng/mL.

### 2.2. Identification of Circulating miRNA Biomarkers Associated to Therapy Response and DSF

Twenty HCC serum samples, collected at the time of diagnosis, were analyzed through the Affymetrix Genechip^®^ mirna 3.0 array build on miRBase release 20. According to the Absent/Present calling of the Affymetrix algorithm 274 mature miRNAs were considered to be expressed in serum of HCC patients and were included in the subsequent analysis. The characterization of present miRNA has already been described elsewhere [20].

In order to identify predictive biomarker for therapy response, we compared the expressed circulatory miRNome of CR vs. PRPD patients (Figure 1, Appendix A). Nine circulating mature miRNAs were significantly different between the two groups. The expression of miR-4454, miR-4492, miR-4722-3p, and miR-4530 were significantly higher in CR compared to PRPD, with miR-4454 having the highest differences between the two groups. On the contrary, the expression of miR-4443, miR-1275, miR-885-3, miR-2116-3p, and miR-4439 were significantly higher in the PRPD group with miR-4443 being the top-scoring miRNA in terms of FC and *p*-value.

This miRNA signature was able to cluster the two groups of patients in CR and PRPD at the time of diagnosis, before any oncological treatment.

### 2.3. Predictive miRNA Biomarkers for Therapy Response

We selected four miRNA candidates (miR-4443, miR-4454, miR-4492, and miR-4530) for the subsequent validation phase with qRT-PCR in 86 HCC serum samples. Patients were grouped according to response to treatment as described in materials and methods. Among the four selected miRNAs, miR-4454, miR-4492, and miR-4530 were significantly and differently expressed between CR and PRPD at T0 (Appendix A), showing an increased fold of change of 1.88, 1.84, and 1.42 in CR, respectively. To further investigate the prognostic significance of the miRNA candidates, we grouped the patients receiving either curative treatments (liver resection and radiofrequency) or TACE.

### 2.4. MiR-4443, miR-4454, and miR-4530 as Predictive miRNA Biomarkers for Therapy Response in Curative Therapies

Considering patients treated with curative therapies (*n* = 41), miR-4443 (*p* = 0.05), miR-4454 (*p* = 0.02), and miR-4530 (*p* = 0.008) were confirmed as significantly and differently expressed between CR and PRPD at T0, with miR-4454 and miR-4530 showing a 2.83 and 2.33 fold of increase in CR, respectively (Figure 2b,c). On the contrary, miR-4443 resulted in a 1.95-fold of down-regulation in CR (Figure 2a). MiR-4492 was not confirmed as significantly associated with the response to curative treatments as there is no significant differential of expression between CR and PRPD (*p* = 0.88). 

To validate the discriminatory potential of the three miRNAs, the area under the curve (AUC) was determined with a ROC analysis (Figure 3). The highest AUC values were obtained for miR-4530 with AUC = 0.77 (95%CI 0.55–0.89), with a sensitivity and specificity of 72% and 72.7%, respectively, at a cut off determined at ≥317.40 (Figure 3c). The corresponding AUC values for miR-4443 and miR-4454 were 0.72 at a cutoff of ≤2.93 (95% CI 0.45–0.87) and > 0.09 (95% CI: 0.49–0.88), respectively (Figure 3a,b). Mir-4454 has the highest sensitivity in distinguishing CR from PRPD before therapy (sens = 79%) (Figure 3b). The combination of the three miRNA showed a greater ability than any individual miRNA to distinguish CR from PRPD with an AUC of 0.84 (95% CI: 0.45–0.87) using an optimal cut-off value of 0.50, which demonstrated a sensitivity of 72% and specificity of 75% (logit model formula: −1.89 + 0.32*miR-4492 + 6.45*miR-4454 − 0.01 × miR-4492 × miR-4492 − 0.75 × miR-4492 × miR-4454 − 0.90 × miR-4454 × miR-4454) (Figure 3d).

### 2.5. Higher Levels of miR-4454 and miR-4530 Predict Longer Survival in HCC Patients

We also analyzed the potential of the three miRNAs (miR-4454, miR-4530 and miR-4443) to predict an overall survival of patients before initiation of therapy (T0). Cut-off values determined by ROC curve analysis were used to compare the survival time of patients with low vs. high miR expression. Kaplan–Meier survival analysis demonstrated that high expression of miR-4454 and miR-4530 were significantly related to better overall survival in patients, with a HR of 2.79 (95% CI: 0.85–2.16, *p* = 0.029) (Figure 4b) and HR of 2.97 (95% CI 1.02–8.62, *p* = 0.011) (Figure 4c), respectively. However, the expression of miR-4443 was not significantly related to overall survival (Figure 4a).

### 2.6. MiR-4454 and miR-4530 are Predictors for Longer DFS for Curative Treatments

Patients receiving curative treatment were then separated into two groups, patients with Disease-Free Survival (DFS) ≤ 6 months and patient with DFS > 6 months. The expression of MiR-4454 and miR-4530 were 2.61 (*p* = 0.03) and 1.93 (*p* = 0.015) times higher in patients with DFS > 6 months, respectivelly (Figure 5a,b). To test the discriminatory potential of the two miRNAs, we calculated the AUC of ROC curve for both miRNAs. Using a cut-off of ΔΔCq ≥ 0.20, miR-4454 can distinguish DFS > 6 months at T0 with a sensitivity and specificity of 67% and 64%, respectively (AUC = 0.73 (95% CI: 0.50–0.86), *p* = 0.005) (Figure 5c). Similarly, at a cut-off of ΔΔCq ≥ 539.50, miR-4530 can distinguish both groups with sensitivity and specificity of 70% and 80%, respectively (AUC = 0.78 (95% CI: 0.56–0.90), *p* = 0.0003) (Figure 5d). However, combination of the two miRNAs showed better performances in distinguishing patients with longer DFS, with a sensitivity increased up to 79% and a specificity of 72%, calculated at a cut-off of >ΔΔCq 0.53 (AUC = 0.81 (95% CI: 0.59–0.82) *p* = 0.008) (Figure 5e). The Logit model formula is: −1.63 + 2.95 × miR-4454 + 0.00 × miR-4530.

### 2.7. MiR-4492 as a Predictive miRNA Biomarker for Therapy Response to TACE

We analyzed the predictive value of the four miRNA candidates in HCC patients receiving TACE. None of the three miRNAs (miR-4443, 4454, and miR-4530), previously able to significantly predict the response to curative treatment, were confirmed as prognostic biomarkers in patients treated with TACE. On the opposite, miR-4492 was 2.67-fold times (*p* = 0.01) higher in CR (*n* = 14) compared to PRPD (*n* = 32) after TACE at T0 (Figure 6a). The discriminatory potential of miR-4492 was measured by calculating the area under the curve (AUC) in a ROC curve analysis. The corresponding AUC value was 0.84 (95% CI: 0.57–0.91) for differentiating CR patients compared to PRPD with sensitivity and specificity of 84.6% and 71%, respectively, using a cut-off of ΔΔCq = 12.60 (Figure 6b).

### 2.8. Proposed Model of Serum miRNAs as Predictive Biomarker in Clinical Setting

Utilizing the significance and specificity of miR-4443, -4454, -4530 to predict therapy response for curative treatments and miR-4492 for TACE, we created a schematic model to translate this result in the clinical setting (Figure 7). Serum samples of HCC patients found to be eligible for curative treatments or TACE according to the current guidelines will be assessed according to the set of significant miRNAs for each treatment. The logit formula of miR-4443, -4454, and -4530 ≥ 0.50 predicts a positive response to curative treatments. Concordantly, high expression of miR-4454 and miR-4530 are significantly related to longer overall survival. Thus, patients with logit formula <0.5 are considered to be poor responders to curative treatments. Similarly, for patients eligible for TACE, ΔΔCq miR-4492 expression at a cut-off ≥ 12.60 will predict a positive response to TACE treatment, while expression level < 12.60 will give a suggestion to a clinician to consider other treatments.

## 3. Discussion

Due to the unsatisfactory outcomes of available HCC treatments, a non-invasive biomarker that predicts therapy response and survival of patients before receiving treatments might be a feasible alternative to help clinicians to apply the right treatment strategy for each patient. In the present study, we investigated the potential of serum miRNAs as a prognostic biomarker after HCC treatment from serum samples taken before therapy (T0) in 86 HCC patients receiving either curative (resection and thermoablation) or TACE. To our knowledge, this is the first study describing the role of miR-4443, miR-4454, miR-4492, and miR-4530 in relation to HCC. In patients receiving curative treatments, high levels of miR-4454 and miR-4530, and low level of miR-4443 were able to predict the response to therapy at T0. ROC curve analysis confirmed the discriminatory potential of the three miRNAs in distinguishing CR from PRPD patients. However, better performances were obtained by the combination of the three miRNAs at T0, suggesting their potential use as non-invasive predictors for the response for curative treatment before therapy initiation.

Interestingly, the high expression of both miR-4454 and miR-4530 at T0 were also associated with longer DFS. This is in line with the previous findings in which high miR-4454 and miR-4530 levels where associated with CR. As expected, patients responding to treatment and with longer DFS have also longer survival time, and the levels of miR-4454 and miR-4530 are able to predict the favorable prognosis. Thus, the different expression profile of miR-4443, miR-4454, and mir-4530 before the treatment, might suggest their potential role as an independent biomarker to predict the success of treatment. Moreover, only patients with high levels of circulating miR-4454 have also longer OS, strengthening the importance of these miRNAs in several aspects of the disease. The observed differences in the expression of the three miRNAs might reflect the different molecular profile of the tumor. Distinct HCC molecular subtypes could express different miRNA levels, which might influence several critical steps in cancer pathways, such as invasion, apoptosis and other mechanisms closely related to proliferation and recurrence. Thus, the low expression of miR-4454 and miR-4530 and high expression of miR-4443 in partial responders might be related to some molecular mechanisms that significantly contribute to the response towards treatments.

The high expression of circulating miR-4454 has been documented in two studies conducted in bladder cancer and melanoma [21,22]. Contrasting evidence regarding miR-4443 were reported in several cancer settings, such as breast and ovarian cancer [23,24,25]. Only one study has associated the high expression of miR-4530 with chemosensitivity in breast cancer [26]. This miRNA directly targets Runt-related transcription factor 2 (*RUNX2)* that interacts with p53 and is correlated with poor clinical outcomes and resistance to anthracylines [26]. Therefore, it is interesting to assess whether miR-4454 and miR-4530 are targeting the same pathway or different pathways but likewise responsible for the poor prognosis and more aggressive phenotype of cancer. 

In patients receiving TACE, we discovered that none of the three miRNAs with prognostic significance for curative treatments confirmed their value for TACE. On the contrary, miR-4492 was significantly associated with CR at T0. The value of this miRNA as a predictive biomarker for response TACE was confirmed by a ROC curve analysis showing an AUC of 0.84 with a sensitivity of 84.6% and specificity of 71%, which is superior compared to the current biomarker, such as AFP, used for predicting responders after TACE [11,27,28]. The only study describing the role of miR-4492 in cancer comes from Lu et al. (2018) in colorectal cancer [29]. It was described that miR-4492 targets forkhead box K1 (*FOXK1*), an oncogene that regulates proliferation and invasion in colorectal cancer cells [29]. However, the role and key mechanism of these targets should be validated in HCC in relation to therapy response towards TACE.

Taken together, this result underlines the specificity of utilizing the appropriate miRNAs as a predictive biomarker of TR for a specific type of therapy. We proposed a schematic model to utilize our panel of biomarker candidates to help clinicians choose a proper treatment, together with the available clinical guidelines (Figure 7). This might be in line with the future goal to apply individualized treatment protocols to every single HCC patients. Here, we identified a panel of novel miRNAs that were never reported as a non-invasive biomarker candidate in the HCC setting but proven to be associated with cancer pathways in other types of cancer. The discriminatory potential for TR, DFS and OS of our miRNAs panel at T0 might strengthen the relevance of miRNAs as circulating biomarkers, supporting the idea of individualized treatment strategies based on risk prediction model to predict the outcomes for each type of therapies. To ensure the reproducibility of these results, a multi-center validation study needs to be conducted in a larger cohort considering that individual variability such as gender, race, or etiology of cancer might also influence the miRNA profile. 

## 4. Materials and Methods 

### 4.1. Patients

Twenty consecutive patients referring to the Liver Center between 2012 and 2017 who were diagnosed with HCC according to the EASL criteria were enrolled for the study. Samples were collected at the time of HCC diagnosis (T0). The clinical and demographic features of the groups are shown in Appendix A.

All the patients provided written informed consent and patient anonymity has been preserved. The investigation was conducted according to the principles expressed in the Declaration of Helsinki. The study was approved by the regional ethical committee (Comitato Etico Regionale Unico FVG, No. 14/2012 ASUITS).

### 4.2. Study Design

This study was organized as follows: the discovery phase (phase 1). Serum samples collected at the time of diagnosis from 20 patients were analyzed through microarray profiling. After therapy patients were categorized, according to modified Response Evaluation Criteria In Solid Tumors (RECIST) as Complete Responders (CR), Partial Responders (PR), or Progressive Disease (PD). Kruskal–Wallis test was used to determine gene expression differences in microarray analysis between CR patients vs. PR + PD patients (PRPD). In the discovery group, a panel of candidate circulating predictive biomarkers for therapy response was selected based on the differential fold of change (cut-off: ± 1.5) and significance level (*p* < 0.05). Validation phase (phase 2). MiRNA candidates selected in the discovery phase were tested by quantitative Real-Time PCR (RT-qPCR) in other 86 HCC serum samples. Patients were divided into two groups consisting of patients receiving curative treatments and patients receiving TACE treatments. mRECIST criteria were used to define the response to treatment. MiRNA expression profiles determined at T0, were then associated with the response to different therapies.

### 4.3. Serum Samples Collection and RNA Isolation

Serum samples were obtained from 6 mL of whole blood collected in Vacuette^®^ serum separating tubes (Greiner Bio-One GmbH, Kremmunster, Austria) and centrifuged at 3500 rpm for 10 min. Supernatants were transferred in 1 mL eppendorf tubes and subsequently frozen at –80 °C for long-term storage.

Small RNAs, were isolated from 300 uL of serum using the miRCURY™ RNA Isolation Kits (Exiqon, Vedbaek, Denmark) and quantified in a Qubit^®^ 2.0 Fluorometer (Thermo Fischer Scientific, Waltham, MA USA) by using the Qubit microRNA Assay Kit (Thermo Fischer Scientific, Waltham, MA, USA) following the manufacturer instructions. The quality of extracted RNAs was assessed with the Agilent Small RNA kit (Agilent Technologies, Santa Clara, CA, USA) by using the 2100 Bioanalyzer Instrument (Agilent Technologies, Santa Clara, CA, USA).

### 4.4. Microarray Profiling and Data Analysis 

We labelled 130ng of purified small RNAs with the FlashTag™ Biotin HSR RNA Labeling Kit (Affymetrix^®^, Thermo Fischer Scientific, Waltham, MA USA) and hybridized on Genechip miRNA 3.0 (Thermo Fischer Scientific, Waltham, MA USA) containing 1734 human mature miRNAs. Array cartridges were processed on an Affymetrix Fluidic Station 450 (Affymetrix^®^, Thermo Fischer Scientific, Waltham, MA USA) and scanned on an Affymetrix GeneChip 3000 7G (Affymetrix^®^, Thermo Fischer Scientific, Waltham, MA USA).

The robust Multichip Analysis (RMA) algorithm was used to derive CEL file probe-level hybridization intensities at the gene expression levels. The Absent/Present calling of the Affymetrix algorithm included in the Affymetrix Transcriptome Analysis Console was used to select the miRNAs considered as “present” in the analyzed samples.

Kruskal–Wallis test was used to determine gene expression differences in microarray analysis. Multiple testing correction was performed with the Bonferroni method and corrected *p*-values were calculated. Heatmaps of the differentially expressed miRNAs, with the pseudocolor scale underneath, were generated by using Mev 4.9.0 software. Unsupervised hierarchical clustering with Pearson’s correlation similarity and average linkage was used to order samples and miRNAs.

### 4.5. qRT-PCR Validation 

Thirty nanograms of microRNAs were reverse-transcribed by using the qScript microRNA cDNA Synthesis Kit (Quantbio, Beverly, MA, USA) according to manufacturer instruction. Samples were analyzed thought RT-qPCR by using the PerfeCTa SYBR^®^ Green SuperMix (Quantbio, Beverly, MA, USA) in a CFX-96 thermal cycler (Bio-Rad Laboratories, Hercules CA, USA) according to manufacturer instructions. All reactions were run in duplicate in a 25 uL reaction. MiRNA primers were purchased from Sigma-Aldrich (Merck KGaA, Darmstadt, Germany). MiR-1280 was used as an endogenous normalizer. Expression levels were calculated by using the 2^−ΔΔCt^ formula.

### 4.6. Statistical Methods

The Mann-Whitney U test was used to compare the differences between the two independent groups. For multiple comparisons, the Kruskal–Wallis test in a One-Way ANOVA procedure was used. The Chi-square test was performed to compare differences in demographic characteristics. The receiver operating characteristic (ROC) curves were plotted to estimate the discriminatory potential of the miRNAs. A hierarchical forward selection with switching one-way logistic analysis was used to estimate the discriminatory potential of the miRNA combination. Survival curves were plotted according to the Kaplan–Meier method. Analyses were performed by using NCSS 11 Software (2016) (NCSS, LLC. Kaysville, UT, USA, ncss.com/software/ncss) and Stata 16.0 (Stata Corporation, College Station, TX, USA).

## 5. Conclusions

We identified the potential of several serum miRNAs to predict patients’ response before the initiation of therapy, specifically for either curative or non-curative treatments. MiR-4454, miR-4530, and miR-4443 were associated with curative treatments (liver resection and radiofrequency ablation), while miR-4493 was associated with non-curative treatment (TACE). Therefore, the utilization of miRNAs as circulating biomarkers might support the idea of individualized treatment strategies based on risk prediction model to predict the outcomes for each type of therapies.

## Figures and Tables

**Figure 1 cancers-12-02810-f001:**
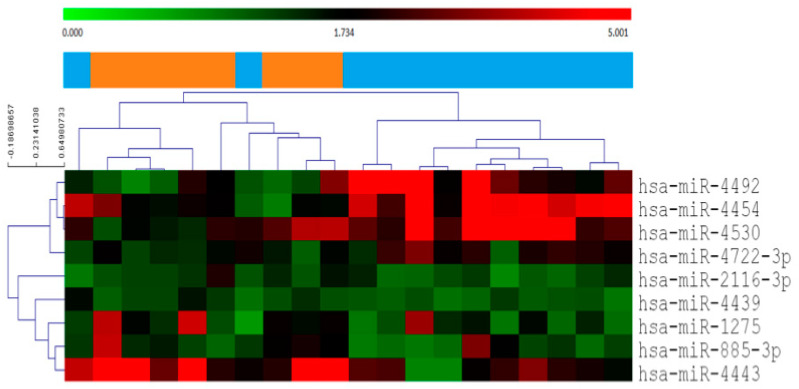
Heatmap with the pseudo-color scale underneath of the differentially expressed miRNAs between complete responder (CR) (blue bar) and partial responder and progressive diseases (PRPD) (orange bar). Unsupervised hierarchical clustering was used to order samples and miRNAs, the log2-transformed microarray signal was considered. The sample tree with optimized leaf-ordering was drawn using Euclidean distances and average linkages for cluster-to-cluster distance. CR = complete responder, PRPD = partial responder and progressive diseases.

**Figure 2 cancers-12-02810-f002:**
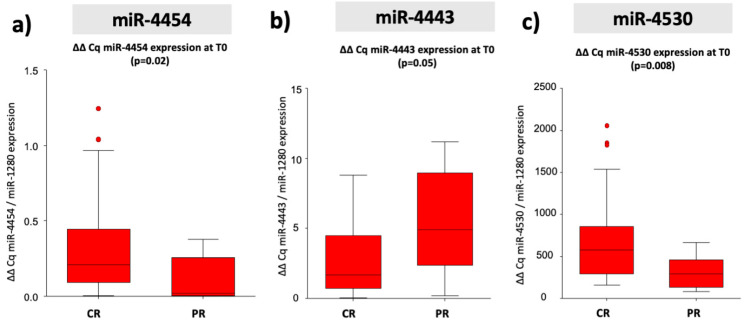
Mean ΔΔCq expression of miR-4443, miR-4454, and miR-4530 in patients underwent curative treatments. MiR-4443 (**a**) resulted in 1.95 fold of down-regulation in CR, while miR-4454 (**b**) and miR-4530 (**c**) showing a 2.83 and 2.33 fold of increase in CR. CR = complete responder, PRPD = partial responder and progressive diseases.

**Figure 3 cancers-12-02810-f003:**
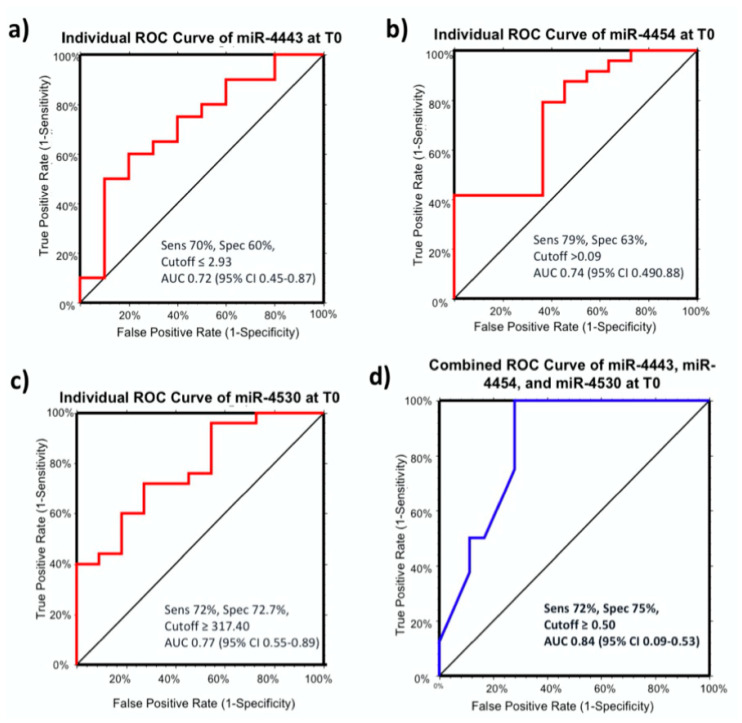
Receiver operating curve (ROC) analysis of miR-4443 (**a**), miR-4454 (**b**), miR-4530 (**c**), and a combination of the three miRNAs (**d**). The combination of the three miRNAs could determine the accuracy on distinguishing CR from PRPD before curative therapy (T0) with an AUC of 0.84 (95% CI 0.45–0.87) using an optimal cut-off value of 0.50, which demonstrated a sensitivity of 72% and a specificity of 75%. CR = complete responder, PRPD = partial responder and progressive diseases.

**Figure 4 cancers-12-02810-f004:**
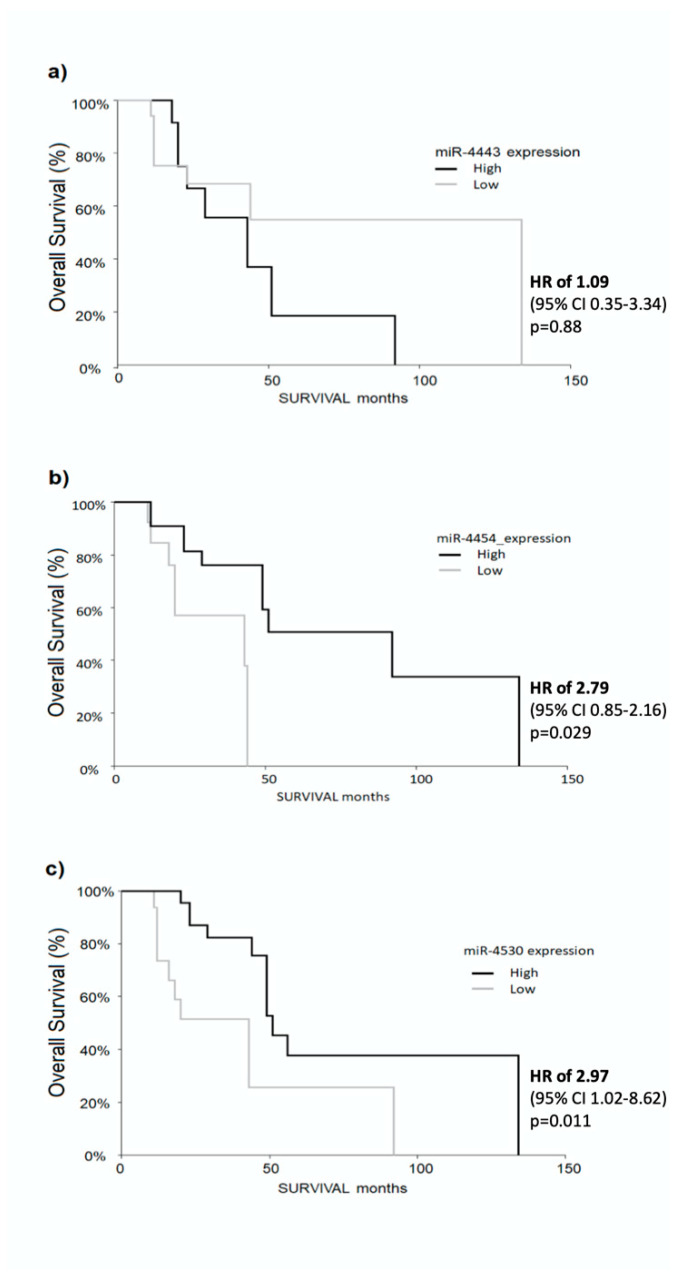
Kaplan–Meier survival analysis by log-rank test for miR-4454 (**a**), miR-4443 (**b**), and miR-4530 (**c**) at T0 for patients receiving curative therapies. It is demonstrated that high expression of miR-4454 and miR-4530 were significantly related to better overall survival in patients, with a hazard ratio (HR) of 2.79 (95% CI: 0.85–2.16, *p* = 0.029) (Figure 4b) and HR of 2.97 (95% CI: 1.02–8.62, *p* = 0.011). HR = hazard ratio.

**Figure 5 cancers-12-02810-f005:**
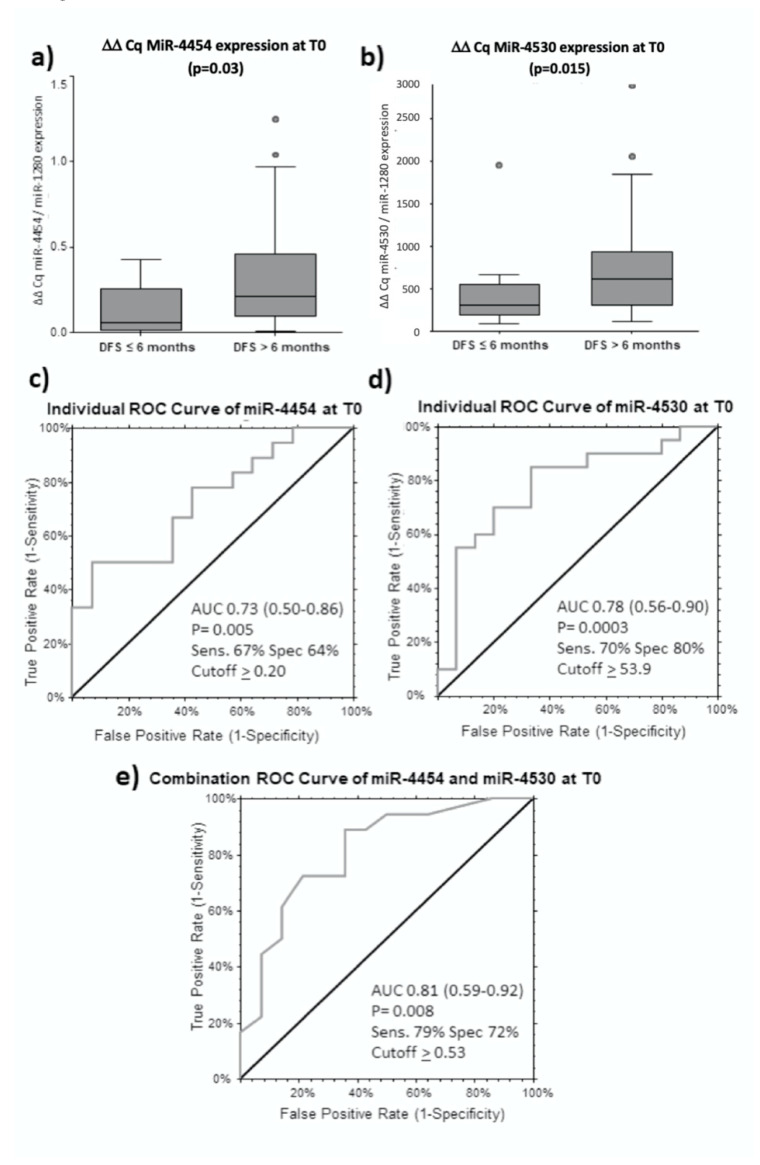
MiR-4454 and miR-4530 are predictors for longer disease-free survival (DFS) for curative treatments. Mean ΔΔCq expression of miR-4454 (**a**) and miR-4530 (**b**) in patients with DFS ≤ 6 months and DFS > 6 months at T0. ROC analysis of miR-4454 (**c**), miR-4530 (**d**) and in combination (**e**).

**Figure 6 cancers-12-02810-f006:**
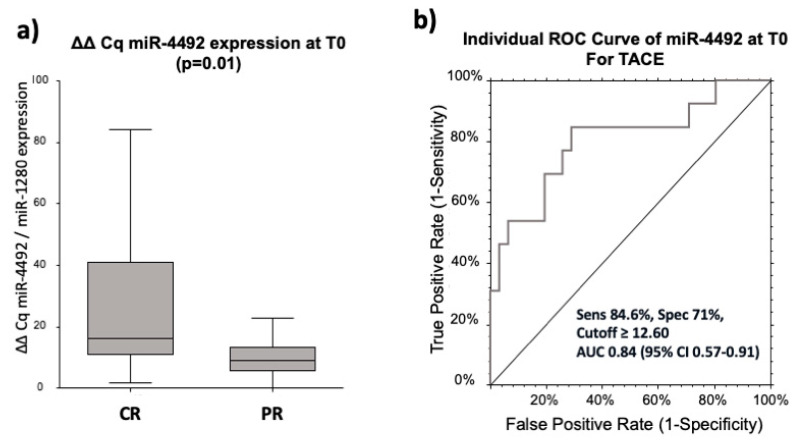
MiR-4492 predicts therapy response to trans-arterial chemoembolization (TACE). Mean ΔΔCq expression of miR-4492 in CR and PRPD in TACE-treated patients (**a**). ROC analysis of miR-4492 (**b**).

**Figure 7 cancers-12-02810-f007:**
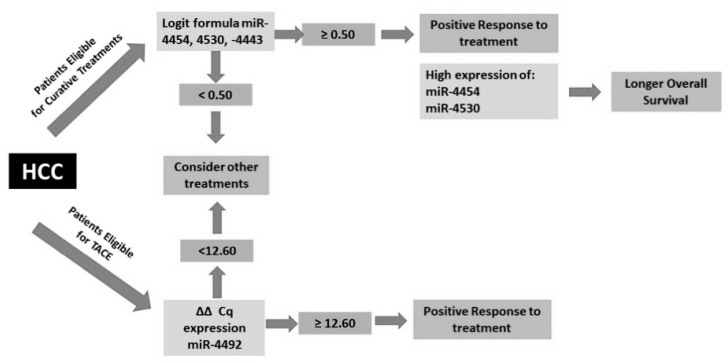
Schematic model to use miR-4443, -4454, -4530 and -4492 as a predictive biomarker to determine HCC treatment in patients. Logit formula value of miR-4443, -4454, and -4530 ≥ 0.50 will predict complete response to curative treatments (resection and radiofrequency ablation) and the Logit formula value of miR-4454 and miR-4530 ≥ 0.53 will predict DFS ≥6 months. ΔΔCq miR-4492 expression at a cut-off ≥12.60 will predict complete response to TACE. DFS = disease-free survival, TACE = trans-arterial chemoembolization.

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
