# Peer review of "Circulatory miRNA as a Biomarker for Therapy Response and Disease-Free Survival in Hepatocellular Carcinoma"

_cancers, 2020, doi:10.3390/cancers12102810_

Round 1
Reviewer 1 Report
Comments to the Author
Author reported that different miRNAs predictors can be used as potential prognostic circulating biomarkers according to the selected treatment for HCC. This report evaluated the usefulness of miRNAs for HCC patients and an interesting perspective.
(1). P231-233, The author described that “The observed differences in the expression of the three miRNAs might reflect the different molecular profile of the tumor” in the manuscript. In patients with curative therapy (operation), how about the gross classification (simple nodular or other than simple nodular) and micro vascular invasion?.
(2). Although author described that number of lesions were not significant difference between curative treatment and TACE in table 1, I think there are significant difference between the CR in TACE group and the PDPR in TACE group about BCLC stage(0A/BC). Is there a relationship mir-4492 depending on the tumor stage in the TACE group?
(2). In the survival curve of Figure 4, number at risk should be added in the Figure.
(3). Figure 8 was not described in this manuscript.
(4). Table1, the number of all patients were 86, but the number of Etiology, Number of lesions, disease scores are incorrected. Moreover, I think the other number are different. Why is the number different?
Reviewer 2 Report
The manuscript by Pratama et al. “Circulatory miRNA as biomarkers for therapy response and disease-free survival in Hepatocellular Carcinoma” reports a circulatory miRNA signature provides prognosis for curative HCC treatment while miR-4492 level is correlated with efficacy of TACE procedure. These findings if substantiated will provide treatment guidance to extend survival for HCC patients with current pathological diagnoses. In particular, for curative treatment, it was shown that similar pathological staging analysis with different miRNA signature could lead to very different outcomes, thus providing significant benefit for selection of effective treatment based on the miRNAs in the circulation. On the other hand, for TACE procedure, the outcome of treatment seems to be preordained with the tumor staging difference (supplementary Table 1, BCLC 0A/BC score), therefore the value of detecting miRNA level is less clear. Significant improvement for this part is necessary, either dropping this portion of the analysis, or redoing the miRNA analysis with patients of similar pathological evaluations. On a minor note, there is one patient difference in terms of the total patient number in Supplementary table 1.
Author Response
Please see the attachment below.

Round 2
Reviewer 1 Report
Thank you for reviewing this paper. I think authors answer all question properly.
This manuscript is worthy to published in this journal.
Reviewer 2 Report
The authors have provided improvement over previous submission.